# Whole-Transcriptome Analysis Identifies Gender Dimorphic Expressions of Mrnas and Non-Coding Rnas in Chinese Soft-Shell Turtle (*Pelodiscus sinensis*)

**DOI:** 10.3390/biology11060834

**Published:** 2022-05-29

**Authors:** Junxian Zhu, Luo Lei, Chen Chen, Yakun Wang, Xiaoli Liu, Lulu Geng, Ruiyang Li, Haigang Chen, Xiaoyou Hong, Lingyun Yu, Chengqing Wei, Wei Li, Xinping Zhu

**Affiliations:** 1Key Laboratory of Tropical & Subtropical Fishery Resource Application & Cultivation of Ministry of Agriculture and Rural Affairs, Pearl River Fisheries Research Institute, Chinese Academy of Fishery Sciences, Guangzhou 510380, China; zhujunxian_1994@163.com (J.Z.); 2019213005@stu.njau.edu.cn (L.L.); chenchen@prfri.ac.cn (C.C.); wangyk@prfri.ac.cn (Y.W.); liuxl@prfri.ac.cn (X.L.); gengll97@163.com (L.G.); a3021893023@163.com (R.L.); zjchenhaigang@prfri.ac.cn (H.C.); hxy@prfri.ac.cn (X.H.); yuly@prfri.ac.cn (L.Y.); zjweichengqing@prfri.ac.cn (C.W.); 2College of Fisheries and Life Science, Shanghai Ocean University, Shanghai 201306, China; 3Wuxi Fisheries College, Nanjing Agricultural University, Wuxi 214081, China

**Keywords:** *Pelodiscus sinensis*, whole-transcriptome sequencing, sex differentiation, non-coding RNAs, ceRNA

## Abstract

**Simple Summary:**

*Pelodiscus sinensis* has significant gender dimorphism in its growth patterns. Nevertheless, its underlying molecular mechanisms have not been elucidated well. Here, a whole-transcriptome analysis of the female and male gonads was performed. Altogether, 7833 DE mRNAs, 619 DE lncRNAs, 231 DE circRNAs, and 520 DE miRNAs were identified and enriched in the pathways related to sex differentiation. Remarkably, the competing endogenous RNA interaction network was constructed, including the key genes associated with sex differentiation. Collectively, this study provides comprehensive data, contributing to the exploration of master genes during sex differentiation and highlighting the potential functions of non-coding RNAs in *P. sinensis* as well as in other turtles.

**Abstract:**

In aquaculture, the Chinese soft-shelled turtle (*Pelodiscus sinensis*) is an economically important species with remarkable gender dimorphism in its growth patterns. However, the underlying molecular mechanisms of this phenomenon have not been elucidated well. Here, we conducted a whole-transcriptome analysis of the female and male gonads of *P. sinensis*. Overall, 7833 DE mRNAs, 619 DE lncRNAs, 231 DE circRNAs, and 520 DE miRNAs were identified. Some “star genes” associated with sex differentiation containing *dmrt1*, *sox9*, and *foxl2* were identified. Additionally, some potential genes linked to sex differentiation, such as *bmp2*, *ran*, and *sox3*, were also isolated in *P. sinensis*. Functional analysis showed that the DE miRNAs and DE ncRNAs were enriched in the pathways related to sex differentiation, including ovarian steroidogenesis, the hippo signaling pathway, and the calcium signaling pathway. Remarkably, a lncRNA/circRNA–miRNA–mRNA interaction network was constructed, containing the key genes associated with sex differentiation, including *fgf9*, *foxl3*, and *dmrta2*. Collectively, we constructed a gender dimorphism profile of the female and male gonads of *P. sinensis*, profoundly contributing to the exploration of the major genes and potential ncRNAs involved in the sex differentiation of *P. sinensis*. More importantly, we highlighted the potential functions of ncRNAs for gene regulation during sex differentiation in *P. sinensis* as well as in other turtles.

## 1. Introduction

Gender dimorphism of morphology and physiology is widely present in the animal kingdom, which contributes to making the world complicated and wonderful [1]. In aquaculture, the outstanding gender dimorphism of growth rate and body size has drawn special attention from scientists [2,3,4], as these factors can be manipulated to achieve higher economic value and broader consumer demand. Consequently, the breeding of a monosexual population has been extensively used in aquaculture species [5,6,7]. The Chinese soft-shell turtle (*Pelodiscus sinensis*) is a vital economic species in aquaculture, with an annual production of over 330,000 tons [8] and outstanding sexual dimorphism in body growth, calipash, and fat content between males and females, with males having higher commercial value than females [9]. Therefore, there is also demand for the single-sex breeding of *P. sinensis*.

Studies on the regulation mechanism of sex determination and sex differentiation in *P. sinensis* are the basis of unisexual breeding. However, achieving consensus on the sex determination mechanism of *P. sinensis* is complicated. Early studies have shown that the sex of *P. sinensis* embryos was affected by the incubation temperature, so researchers inferred that the *P. sinensis* had temperature-dependent sex determination (TSD) [10,11,12]. With the discovery of the ZZ/ZW micro-chromosome, genetic sex determination (GSD) was verified as the sex-determination system of *P. sinensis* [13,14]. Additionally, via molecular cytogenetics and incubation experiments, TSD was ruled out in *P. sinensis* [15]. Currently, although researchers have conducted relevant studies and achieved some success [16,17], they are not sufficient to completely reveal the mechanisms of sex determination and sex differentiation in *P. sinensi**s*. Accordingly, massive studies on molecular mechanisms, particularly the genes and the regulatory pathways, could promote the mechanistic comprehension of sex differentiation and/or sex determination in *P. sinensis*.

The regulation of sex determination and sex differentiation is complex and diverse, and the master genes play a key role. Sex-determining genes (SDGs) drive the sex differentiation of bipotential gonads into the ovaries or the testes [18]. The sex-determining region Y (*Sry*) gene [19] and the W-linked DM-domain (*Dm-w*) gene [20] were the first SDGs identified in the XX/XY system and ZZ/ZW system, respectively. Subsequently, other SDGs and candidates, including *Zglp1* [21] and *Sox9* [22] in mammals, *Dmy* [23] and *Foxl3* [24] in fishes, *Dmrt1* [25] and *Dsx* [26] in birds, and *Dmrt1* [16,27] and *Amh* [28] in reptiles, were cognized in GSD species. Therefore, more information about these candidates or their homologs should be provided to help us understand their potential roles in the sex differentiation of *P. sinensis*.

Compared to DNA with coding genes, mammalian genomes have large amounts of non-coding DNAs, known as “junk DNA” or “genomic dark matter”, which can be transcribed into non-coding RNAs (ncRNAs) containing long ncRNAs (lncRNAs), microRNAs (miRNAs), and circular RNAs (circRNAs), and although most of them do not encode proteins, they are involved in multiple biological processes [29,30]. More recently, studies have reported that miR-124 repressed the expression of *Sox9* in mice (*Mus musculus*) ovaries, indicating a major role during ovarian differentiation [31]. In addition, in chicken (*Gallus gallus*) male hypermethylation (MHM), a Z sex chromosome-linked locus adjacent to *dmrt1* was methylated and transcriptionally silent in male cells (ZZ), while it was hypomethylated in female cells (ZW) and then transcribed a long non-coding RNA (MHM). Then, injecting the expression plasmids of MHM into adult chicken testes could inhibit *Dmrt1* expression [32]. Moreover, a circular *Sry* transcript was produced in the testes of adult mice [33,34] and functioned as an miR-138 sponge, resulting in a more than tenfold enrichment of AGO2 (Argonaute 2) [35]. The roles of mRNAs and non-coding RNAs in sex differentiation are not independent of each other, and the theory of competing endogenous RNAs (ceRNA) unifies them as a system. Briefly, the lncRNAs and circRNAs function as a sponge to competitively adsorb miRNAs, thereby promoting mRNA expression [36,37]. Despite the fact that the studies about non-coding RNA are a popular topic in the field of sex differentiation, to our knowledge, studies on *P. sinensis* and turtles, especially the interaction between ncRNAs and mRNAs, are far from adequate.

In the present study, we conducted a transcriptome analysis on the ovaries and testes of *P. sinensis* using Illumina RNA Sequencing (RNA-Seq) technology, established differentially expressed (DE) profiles, and analyzed the pathways of DE mRNAs, lncRNAs, miRNAs, and circRNAs. Furthermore, the correlation networks of lncRNA/circRNA–miRNA–mRNA were constructed to disclose the interactions among them. The relative expressions of candidate mRNAs, lncRNAs, miRNAs, and circRNAs were validated using quantitative reverse-transcription PCR (qRT-PCR). Collectively, the comprehensive data contribute to the exploration of master genes during the sex-differentiation process and highlight the potential functions of non-coding RNAs in *P. sinensis* as well as in other turtles.

## 2. Materials and Methods

### 2.1. P. sinensis Collection and Total RNA Isolation

The Animal Care and Ethics Committee of the Pearl River Fisheries Research Institute, Chinese Academy of Fishery Sciences approved this research, and all experimental protocols and methods were performed in accordance with the relevant guidelines and regulations. A total of 18 healthy 6-month-old *P*. *sinensis* subjects, 9 males and 9 females, were collected and sacrificed humanely. Testes and ovaries were obtained and immediately frozen in liquid nitrogen for storage at −80 °C for RNA isolation. Total RNA was isolated using TRIzol reagent (Ambion, Carlsbad, CA, USA). The consistency and purity were tested by the NanoQ™ (Thermo Scientific, Madison, WI, USA), and the quality was evaluated using an Agilent 2100 bioanalyzer (Agilent Technologies, Santa Clara, CA, USA). Three biological replicates were obtained from the 9 males and 9 females, which were pooled with 3 turtles, respectively.

### 2.2. Library Construction and Sequencing

High-quality RNAs were used for library construction. The mRNAs were first enriched using oligo (dT) magnetic beads. The rRNAs were removed using a Ribo-Zero™ rRNA Removal kit (Epicenter, Madison, WI, USA). The rRNA-depleted RNAs were fragmented using fragmentation buffer and then the first-strand cDNA was synthesized using random hexamer primers. Subsequently, the second-strand cDNA was progressed, adding buffer, dNTPs, RNase H, and DNA polymerase I. The double-stranded cDNA was converted into blunt ends and 3′ ends were adenylated and then purified by AMPure XP (Beckman Coulter, Brea, CA, USA) beads to select insert fragments. After PCR and ligation of the sequencing adapters, the final library was estimated with the Agilent Bioanalyzer 2100 (Agilent Technologies, Santa Clara, CA, USA) and sequenced at Vazyme Biotech Co., Ltd. (Nanjing, China) using an Illumina HiSeq 4000™ System (Illumina, San Diego, CA, USA).

### 2.3. Quality Control and Transcriptome Assembly

The reads, including adapters, > 10% of ploy-N, and those of low quality ( > 50% of the bases had quality scores of ≤ 5), were removed to obtain clean data. Meanwhile, the Q20, Q30, and GC content of the clean data were calculated and used for further bioinformatics analysis. The clean reads from each library were mapped to the reference genome of *P*. *sinensis* (PRJNA221645, NCBI, accessed on 10 January 2022) using Hisat2 v2.1.0 [38]. The mapped reads were assembled with StringTie v1.3.5 [39] and then annotated by the gffcompare program.

### 2.4. Differential Expression Analysis and Functional Annotation

For each library, the fragments per kilobase per million reads (FPKM) and the transcripts per kilobase of exon model per million mapped reads (TPM) scores were calculated using StringTie v1.3.5 [39]. The differentially expressed levels of mRNAs and ncRNAs were evaluated using edgeR (accessed on 13 January 2022) [40]. The critical values of DE mRNAs and DE ncRNAs were given as a false discovery rate (FDR) of <0.05 and an absolute value of the log2 (fold change) of > 2. The Gene Ontology project (GO, http://www.geneontology.org, accessed on 13 January 2022) and Kyoto Encyclopedia of Genes and Genomes (KEGG, https://www.kegg.jp, accessed on 13 January 2022) were used for the functional annotation of DE mRNAs and DE ncRNAs. A *p* value of < 0.05 was considered significantly enriched in GO terms and KEGG pathway analyses.

### 2.5. Validation of Candidate mRNAs and ncRNAs by qRT-PCR

Eight of the DE mRNAs, lncRNA, and circRNAs, and four of the DE miRNAs were randomly selected to validate the RNA-seq data using qRT-PCR. All primers used in this study are provided in Appendix A. The reaction program of the qRT-PCR was set as follows: 95 °C pre-denaturation for 10 min; 95 °C for 15 s, 60 °C for 20 s, and 72 °C for 20 s for 40 total cycles. The expression levels of the DE mRNAs, lncRNAs, and circRNAs were normalized with the ef1α gene [41], while the U6 gene [42] was applied for the normalization of the DE miRNAs. The relative expressions of the DE mRNAs and DE ncRNAs were calculated using the 2^−∆∆Ct^ method [43]. All data are shown as the means ± standard error of the mean (SEM) and were analyzed using ANOVA (accessed on 6 March 2022). *p* < 0.05 was considered to be significantly different.

### 2.6. Construction of the ceRNA Interaction Network

To better elucidate the potential role of DE ncRNAs in the sex differentiation of *P. sinensis*, a lncRNA/circRNA–miRNA–mRNA interaction network was constructed. miRanda software (Memorial Sloan Kettering Cancer Center, New York, NY, USA) was used to forecast the interaction relationships of the mRNAs, lncRNAs, and circRNAs with miRNAs, and Cytoscape 3.2 (National Institute of General Medical Sciences, Bethesda, MD, USA) was applied for visualization.

## 3. Results

### 3.1. Overview of Transcriptome Sequencing

After removing low-quality reads, we obtained the totally clean reads of the mRNAs, lncRNAs, and circRNAs, ranging from 42,487,292 to 46,871,481, and the miRNAs ranging from 9,765,869 to 14,021,878 via the whole-transcriptome analysis of the early female and male gonads in *P. sinensis*. The GC contents of the mRNAs, lncRNAs, and circRNAs were between 45.70 and 47.94%, and those of the miRNAs were between 45.99 and 50.54%. The Q30 bases of the mRNAs, lncRNAs, and circRNAs were greater than 93.98%, and those of the miRNAs were greater than 97.76% (Appendix A). The results of principal component analysis (PCA) showed good similarity among the three biological replicates (Appendix A). The length distributions of the mRNAs and ncRNAs are also provided in Appendix A.

### 3.2. Identification of DE mRNAs and ncRNAs

A volcano plot and heat map were used to exhibit the differentially expressed results (fold change of ≥ 2 and FDR of < 0.05) of the mRNAs and ncRNAs. In total, 7833 DE mRNAs were identified, of which 3192 were up-regulated and 4641 were down-regulated (Figure 1A, Appendix A); 619 DE lncRNAs were identified, of which 385 were up-regulated and 234 were down-regulated (Figure 1C, Appendix A); 231 DE circRNAs were identified, of which 39 were up-regulated and 192 were down-regulated (Figure 1E, Appendix A); and 520 DE miRNAs were identified, of which 219 were up-regulated and 301 were down-regulated (Figure 1G, Appendix A). Likewise, the hierarchical clustering of the DE mRNAs and DE ncRNAs was divided into two distinct clusters, revealing remarkable differences between the ovaries and testes of *P. sinensis* (Figure 1B,D,F,H).

### 3.3. GO and KEGG Analysis of DE mRNAs and ncRNAs

GO and KEGG analyses were applied to annotate the biological functions and enriched pathways of the DE mRNAs and ncRNAs. GO terms were mainly classified into three categories, including biological process, cellular component, and molecular function. For the DE mRNAs and ncRNAs, 7833 DE mRNAs, 619 DE lncRNAs, 231 DE circRNAs, and 520 DE miRNAs were used for GO annotation, respectively (Appendix A), and all of them were mainly enriched in the cellular process of the biological process, the cellular anatomical entity of the cellular component, and binding of the molecular function (Figure 2A,C,E,G). In addition, a total of 25 KEGG signaling pathways of DE mRNAs, 37 KEGG signaling pathways of DE lncRNAs, 16 KEGG signaling pathways of DE circRNAs, and 30 KEGG signaling pathways of DE miRNAs were significantly enriched, respectively (Appendix A). The top three significantly enriched KEGG signaling pathways were related to the neuroactive ligand-receptor interaction, the olfactory transduction, and the calcium signaling pathway in DE mRNAs (Figure 2B); the Ras signaling pathway, the rap1 signaling pathway, and the regulation of actin cytoskeleton in DE lncRNAs (Figure 2D); the phosphatidylinositol signaling system, the glycerolipid metabolism, and the hippo signaling pathway in DE circRNAs (Figure 2F); and the neuroactive ligand-receptor interaction, the Ras signaling pathway, and the regulation of actin cytoskeleton in DE miRNAs (Figure 2H). The KEGG signaling pathways associated with sex differentiation were also annotated, such as the ovarian steroidogenesis, the hippo signaling pathway, and the calcium signaling pathway.

### 3.4. Validation of DE mRNAs and ncRNAs

Eight DE mRNAs (*foxl2*, *bmp2*, *ran*, *sox3*, *dmrt1*, *dkkl1*, *spo11*, and *sox9*), lncRNA (lncRNA_21915, lncRNA_34729, lncRNA_27020, lncRNA_52064, lncRNA_32111, lncRNA_16229, lncRNA_15672, and lncRNA_55405), circRNAs (circRNA_1083, circRNA_1243, circRNA_1021, circRNA_546, circRNA_464, circRNA_389, circRNA_823, and circRNA_668), and four DE miRNAs (Chr1_880, Chr20_20074, Chr9_14462, and Chr22_21171) were chosen for qRT-PCR analysis to evaluate the reliability of the RNA-seq data presented with the TPM value [44]. The results show that the data of the qRT-PCR (Figure 3B,D,F,H) were consistent with the RNA-seq data (Figure 3A,C,E,G), affirming the accuracy of the RNA-seq data and indicating the existence of sexually dimorphic expression profiles between the female and male gonads of *P. sinensis*.

### 3.5. Construction of the lncRNA/circRNA–miRNA–mRNA Network

The interaction network was predicted using miRanda software with the default parameters [45]. Fold changes greater than 2 and less than -2 indicate up- and down-regulation, and q value < 0.05. Altogether, 131 DE lncRNAs, 40 DE miRNAs, and 1690 DE mRNAs were used to establish the lncRNA–miRNA–mRNA interaction network (Figure 4A, Appendix A), and 17 DE circRNAs, 16 DE miRNAs, and 749 DE mRNAs were applied to build the circRNA–miRNA–mRNA interaction network (Figure 4B, Appendix A). The circles, triangles, and squares represent the miRNAs, lncRNAs or circRNAs, and mRNAs, respectively. Red indicates up-regulation and green indicates down-regulation. The gray lines illustrated their interactions with each other. Remarkably, we identified some pivotal genes, such as *fgf9*, *foxl3*, and *dmrta2*, in the two interaction networks that participate in sex differentiation. *Fgf9* could be regulated with lncRNA_3086, lncRNA_34437, and lncRNA_42520 by the miRNAs of Chr19_19503, Chr19_19503, and Chr1_3952, respectively. Additionally, *Fgf9* could also be regulated with circRNA_302 and circRNA_1030 through the miRNAs of Chr21_20215. *Foxl3* could be regulated with six lncRNAs (lncRNA_21044, lncRNA_35038, lncRNA_37410, lncRNA_44739 lncRNA_47846, and lncRNA_49172), and all by an miRNA of Chr24_22338. *Dmrta2* was regulated with lncRNA_62580 via the miRNAs of Chr6_12669, Chr1_3133, and Chr1_3134 (Figure 4, Appendix A).

## 4. Discussion

In this study, we performed a whole-transcriptome analysis of the early female and male gonads in *P. sinensis*. The results of the quality of each library and the verification of qRT-PCT demonstrated that we established a high-quality and reliable transcriptome database.

Subsequently, we identified 7833 DE mRNAs. Undoubtedly, the “star genes” associated with sex differentiation were identified, including *dmrt1*, *foxl2*, and *sox9*. *Dmrt1* was the first gene identified to be required for the male sex differentiation of *P. sinensis* and exhibited a sex-dimorphic expression pattern throughout the embryonic period. An RNA interference experiment showed that the knockdown of *dmrt1* in male embryos of *P. sinensis* led to sex reversal [16]. Moreover, *Dmrt1* was also the first master gene identified for sex determination in *Trachemys scripta elegans* (a TSD species), and *dmrt1* was directly promoted for transcription by KDM6B to eliminate the trimethylation of H3K27 near its promoter, thereby opening the pathway of male sex differentiation [27]. In the XY system, *sox9* was a downstream target [46] of *sry*, which was located on the Y chromosome, initiating the development of a bipotential primordial gonad in the testes [47]. *sox9* is essential for male sex differentiation in the red-eared slider turtle [48]. *Foxl2* is a member of the Wnt pathway and was the downstream target of *wnt4* [49]. A loss of *foxl2* suppressed *sox9* expression, resulting in a partial reversal of the ovaries [50].

Notably, some novel genes in *P. sinensis* related to sex differentiation were also identified, including *bmp2*, *sox3*, *dkkl1*, *spo11*, and *ran*. *Zglp1* was a downstream effector of *bmp2*, activating the oogenic program and determining the oogenic fate [21]. *Bmp2* and retinoic acid synergistically affirmed the female pathway [51] and responded to Wnt4, a classical feminizing signal [52]. The deletion of *sox9* induced the expression of *bmp2* in XY gonads [53,54]. In addition, *bmp2* presented gender dimorphic expression in *White leghorns* gonads, and was upregulated and hypermethylated in the ovary [55,56]. *Sox3* was the evolutionary ancestor of *sry* and could functionally substitute *sry*, opening the male pathway [57]. In *Oryzias dancena*, the functional deficiency of *sox3* resulted in sex reversal, and *sox3* started male sex differentiation via upregulating the expression of *gsdf* [58]. In *Epinephelus coioides*, a hermaphroditic fish, the continuous expression of *sox3* developed the primordial germ cells toward oogonia; however, when *sox3* ceased expression, the primordial germ cells developed toward spermatogonia [59]. In *Rana rugosa*, *sox3* activated the transcription of *cyp19* by binding to the promoter region, driving the development of gonads into the ovaries [60]. *Dkkl1* was only detected in the testes of mice and humans [61,62], and was mainly distributed in mice spermatocytes and the acrosomes of sperms [63], and in human spermatocytes and round spermatids [61] involved in meiosis and spermatogenesis. Moreover, *dkkl1* could decrease the transcriptional activity of *sf1*, a key sex differentiation gene [64], thereby inhibiting the expression of *cyp17* and *cyp11* and reducing testosterone synthesis in Leydig cells [65]. *Spo11*, a meiosis-specific protein, initiated meiotic recombination in multiple species [66] and the knockout of *spo11* caused meiotic arrest, the apoptosis of spermatocytes during early prophase, and follicle deficiency [67]. *Spo11* was also identified as a male-biased gene in *Danio rerio* [68] and *Anguilla japonica* [69]. Males of *spo11* knockout were completely sterile, but females were fertile [68]. *Ran* could function as an androgen receptor coactivator and weak coactivation of *ran* might result in partial androgen insensitivity [70,71]. The nuclear entry pathway of the sex-determining factor *sry* was demonstrated to require the canonic ran-dependent pathway [72,73]. Likewise, the significant differences of these novel genes in the sex-dimorphic expression patterns of female and male gonads may suggest that their potential function in the sex differentiation of *P. sinensis* has not yet been annotated.

Recently, studies have shown that ncRNAs are extensively involved in the process of sex differentiation [31,32,33,34,35]. Thus, in all of the RNA-seq data, we identified 619 DE lncRNAs, 231 DE circRNAs, and 520 DE miRNAs. The DE mRNAs and DE ncRNAs were enriched in the sex differentiation-associated pathways. Estradiol (E2) is an essential steroid hormone in the ovaries, which has been confirmed to participate in the sex differentiation of turtles. E2 treatment caused the ovarian differentiation of female embryos and sex reversal of male embryos in *P. sinensis* [74], resulting in a rapid up-regulation of *cyp19a1* and completing the sex reversal of male embryos in *Mauremys reevesii* [75]. *Yap* and *wwtr1* were two downstream effectors of the hippo signaling pathway that were inactivated, leading to the up-regulation of female sex-differentiation genes, such as *foxl2* and *wnt4*, and the down-regulation of male sex-differentiation genes, such as *dmrt1* and *sox9* [76]. The calcium signaling pathway was broadly connected to diverse biological processes [77]. Recently, a study found that high temperatures promoted an influx in Ca^2+^ in the somatic cells of gonads, thereby activating pSTAT3 and inhibiting Kdm6b and then Dmrt1 expression, and then ultimately turning on the female pathway [78].

Meanwhile, based on RNA-seq data, we constructed a lncRNA/circRNA–miRNA–mRNA interaction network. Particularly, sex differentiation-related genes *fgf9*, *foxl3*, and *dmrta2* were predicted to be regulated by three pairs of lncRNA–miRNA and two pairs of circRNA–miRNA, and six pairs of lncRNA–miRNA and three pairs of lncRNA–miRNA, respectively. Previous studies have suggested that *fgf9* is a downstream target of *sry*, sufficiently inducing the male sex fate in XX individuals [79]. The lncRNA *LOC105611671* was an upstream regulator of *fgf9* and regulated *fgf9* expression by targeting oar-miR-26a, thereby enhancing testicular steroidogenesis in Hu sheep [80]. A member of the fibroblast growth factor (FGF) family could interact with miRNA-541 to suppress androgen receptor signals in prostate cancer [81]. In *Oryzias latipes*, *foxl3* expressed in germ cells took part in sex determination, and the knockdown of *foxl3* produced functional sperms in females [24]. A study showed that miR-9 targeted the 3’ untranslated region of *foxl3*, promoting oocyte degeneration and meiosis in *Monopterus albus*. miR-430 directly interacted with a member of the forkhead box (FOX) family and repressed its expression [82]. In *Scophthalmus maximus*, *dmrta2* was also identified as a candidate gene related to sex differentiation [83]. A member of the doublesex and mab-3-related transcription factor (DMRT) family was transcriptionally promoted by a transcription factor of the lncRNA DMRT2-AS (referred to as *dmrt2* antisense), participating in the sex differentiation of *Cynoglossus semilaevis* [84]. Analogously, these ncRNAs, which were predicted to regulate the sex differentiation-related genes, may play a vital role in the sex differentiation process of *P. sinensis* and should be further studied.

## 5. Conclusions

In summary, we have provided complete transcriptome data that reveal a gender-dimorphic expression profile in the female and male gonads of *P. sinensis*, which contributes to exploring the master genes and the potential ncRNAs closely related to sex differentiation, as well as understanding the relationship between genes and ncRNAs in the regulatory mechanism of sex differentiation in *P. sinensis*.

## Figures and Tables

**Figure 1 biology-11-00834-f001:**
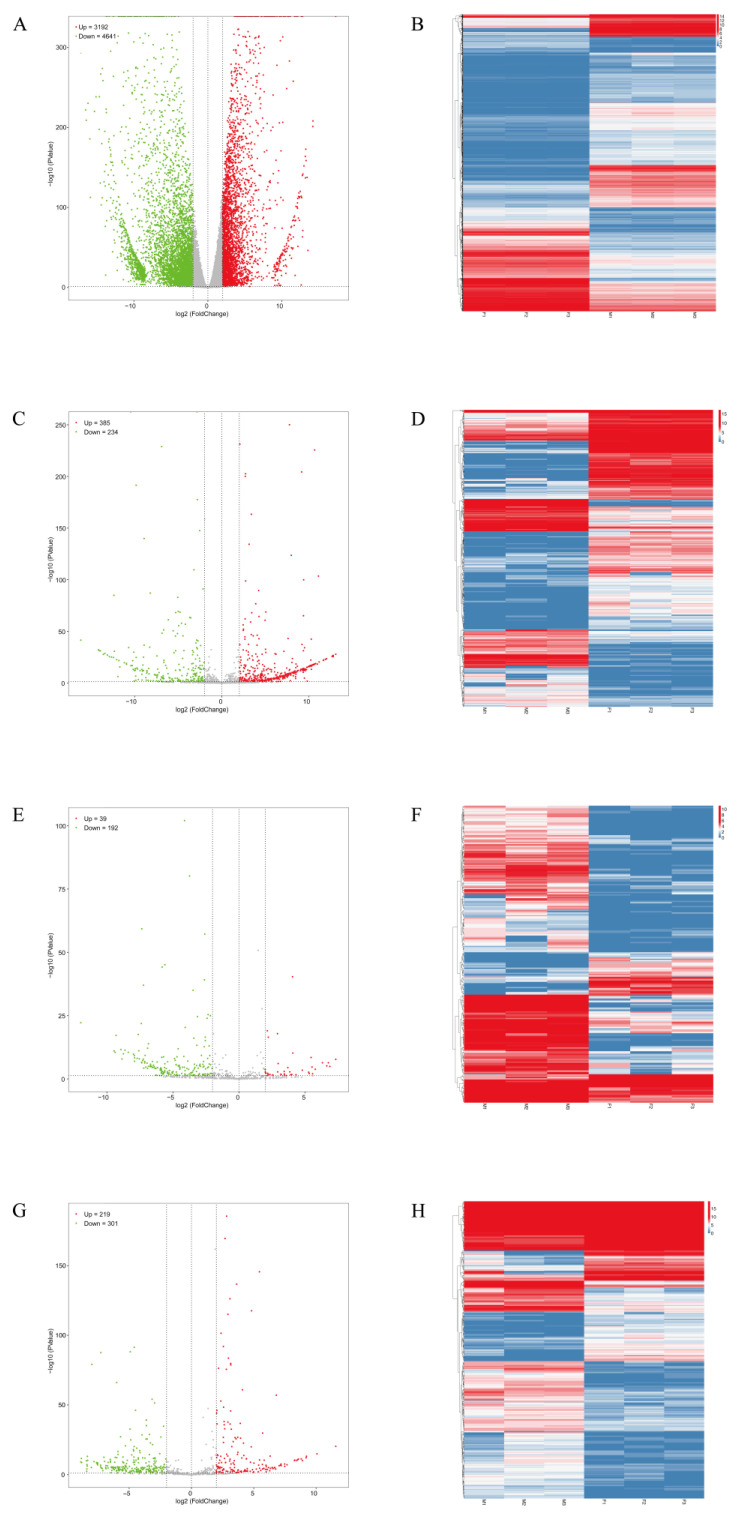
The volcano plots and heat maps of DE mRNAs and ncRNAs. (**A**,**C**,**E**,**G**) indicate the volcano plots of DE mRNAs, DE lncRNAs, DE circRNAs, and DE miRNAs, respectively. (**B**,**D**,**F**,**H**) show the heat maps of DE mRNAs, DE lncRNAs, DE circRNAs, and DE miRNAs, respectively.

**Figure 2 biology-11-00834-f002:**
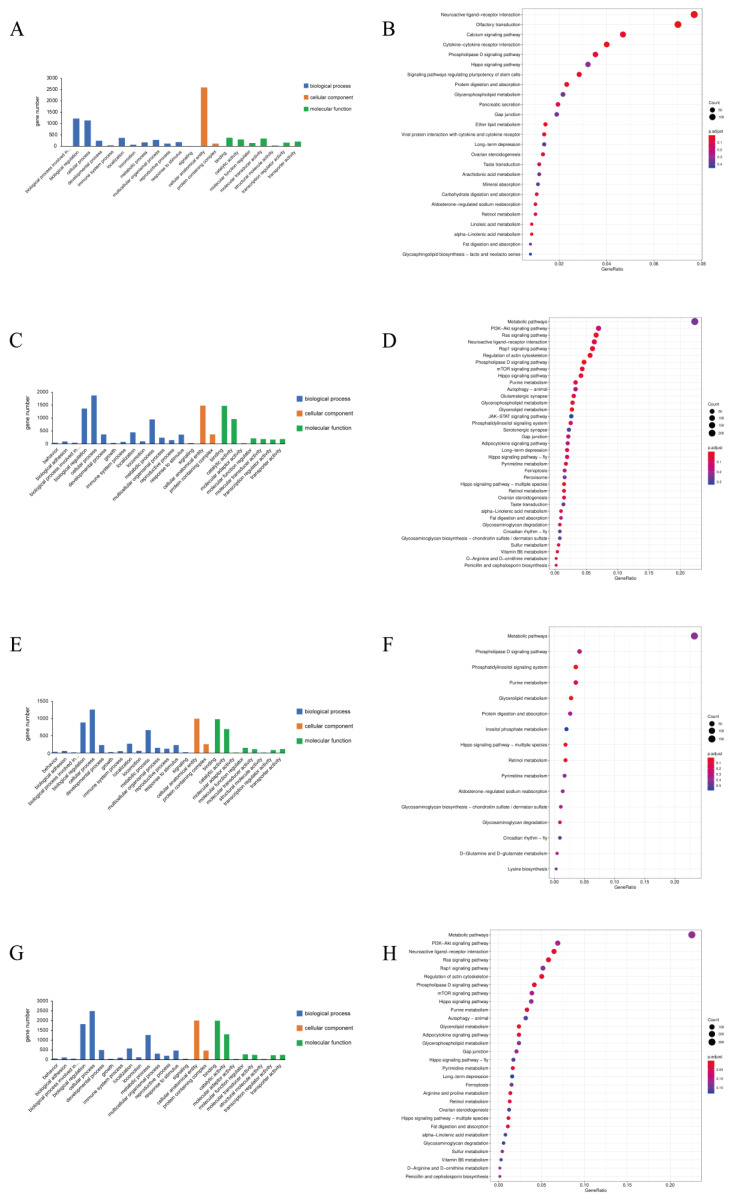
Analyses of functions and pathways of DE mRNAs and ncRNAs. (**A**,**C**,**E**,**G**) signify the GO terms of DE mRNAs, DE lncRNAs, DE circRNAs, and DE miRNAs, respectively. (**B**,**D**,**F**,**H**) illustrate the KEGG signaling pathways of DE mRNAs, DE lncRNAs, DE circRNAs, and DE miRNAs, respectively.

**Figure 3 biology-11-00834-f003:**
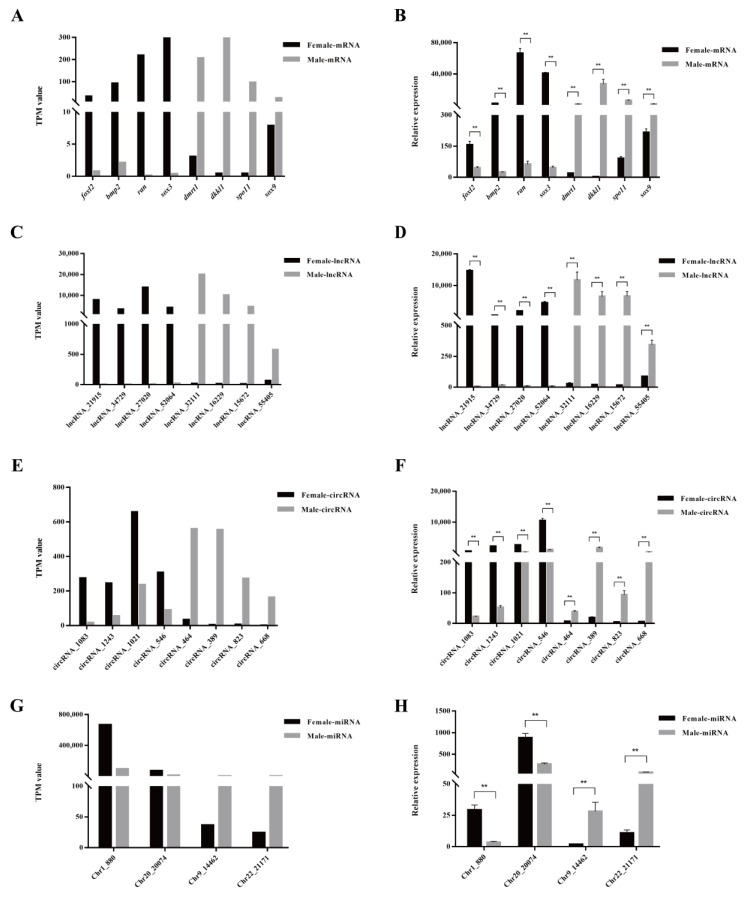
Validation of RNA-seq data by qRT-PCR. (**A**,**C**,**E**,**F**) show the tendencies of mRNAs, lncRNAs, circRNAs, and miRNAs, respectively, in RNA-seq data using the TPM value. (**B**,**D**,**F**,**H**) represent the relative expressions of candidate mRNAs, lncRNAs, circRNAs, and miRNAs, respectively, by qRT-PCR. The TPM value is represented as the means, while the data of the relative expressions shown are the means ± SEM. The asterisk indicates significant differences. ** *p* < 0.01.

**Figure 4 biology-11-00834-f004:**
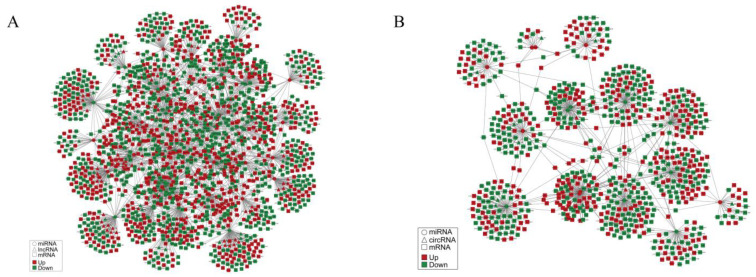
(**A**) The interaction network of lncRNA/circRNA–miRNA–mRNA, and (**B**) the interaction network of circRNA–miRNA–mRNA.

## Data Availability

The raw sequence reads used for analysis in this study can be obtained from the Sequence Read Archive (SRA) under BioProject PRJNA838782. All data are available upon request from the corresponding author.

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
