# Peer review of "Whole-Transcriptome Analysis Identifies Gender Dimorphic Expressions of Mrnas and Non-Coding Rnas in Chinese Soft-Shell Turtle (*Pelodiscus sinensis*)"

_biology, 2022, doi:10.3390/biology11060834_

Round 1
Reviewer 1 Report
The manuscript of the title “Whole-transcriptome analysis identifies gender dimorphic expression of mRNAs and non-coding RNAs in Chinese soft-shell turtle (Pelodiscus sinensis)” provides several novel genes involved in sex determination which could be beneficial for aquaculture work that several species require mono sex aquaculture. However, there are several points that need to improve in the manuscript as follows.
What is “And” in the list of author-name?
Why some texts were bold (e.g., line 14, line 39-40, line 44-59) whether it is technical problem or intentional. Please correct them.
Line 41 Reference for statistics of the annual production of aquaculture products should be from the original sources like the Department of Fisheries or FAO. The author may try the software “FishStatJ” to acquire this information. It would provide you with the better recent information on soft-shell turtle production rather than citing the previous article.
Line 60-61 I didn’t agree with the authors to mention that “whether these candidates or their homologues are gender-dimorphic ex-60 pressed and present the crucial effects in sex differentiation of P. sinensis, are not well clarified.” I think at least Dmrt1 was very well characterized and proven its function in the sex determination of P. sinensis (Reference no. 25). How it was not well clarified. The author should say the way to add more information rather than saying the way of no information available.
Also, the flow of the introduction is not well. Please consider rewriting or rearranging the story for smoother reading.
Line 69 MHM for the first time present, it needs to be clearer to say that MHM is a Z sex chromosome-linked locus adjacent to the DMRT1 gene.
Line 96 Sample size is too small to validate using qPCR primers. Since the author has already developed the primers for qPCR, the author should perform additional samples to assure that these genes were sex differences in other populations as well.
Line 112 Because most Illumina sequencing was performed by outsourcing or company, normally we mentioned the location of sequencing or Illumina equipment. In this case, the author may mention again if the sequencing was performed in your institute (provide the location of sequencing).
All figures were hard to see what it is. The reviewer sees solely that there are differences but is not sure what it is. Please improve the quality of your figures and provide sufficient information into each figure legend.
Line 208 The author should provide the correlation coefficient of the results showing that the data of qRT-PCR were consistent with the RNA-seq data.
In the discussion part, several lines were redundant to the previous session (e.g., abstract, introduction, and result). For example, line 270-274 was redundant to line 178. Please avoid redundant information.
The reviewer suggests the author discuss more novel genes related to sex determination e.g. bmp2, sox3, dkkl1, spo11, and ran in other species. At the current, it was too little to be discussed. The reviewer cannot see how these genes will be important to sex differentiation. The same to discuss ncRNA, circRNA, miRNA, and ceRNA including their interactions. The author should focus to discuss your results and how they could be important for sex differentiation and should be further studied.
Author Response
Dear Editors:
On behalf of my co-authors, we thank you very much for giving us an opportunity to revise our manuscript (ID: biology-1721182), we appreciate editor and reviewers very much for their positive and constructive comments and suggestions on our manuscript. We have studied reviewer’s comments carefully and have made revision. We have tried our best to revise our manuscript according to the comments. Attached please find the revised version, which we would like to submit for your kind consideration.
We would like to express our great appreciation to you and reviewers for comments on our paper. Looking forward to hearing from you.
Thank you and best regards.
Yours sincerely,
Corresponding author: Xinping Zhu
E-mail: zhuxinping_1964@163.com
Reviewer #1:
Comments and Suggestions for Authors
The manuscript of the title “Whole-transcriptome analysis identifies gender dimorphic expression of mRNAs and non-coding RNAs in Chinese soft-shell turtle (Pelodiscus sinensis)” provides several novel genes involved in sex determination which could be beneficial for aquaculture work that several species require mono sex aquaculture. However, there are several points that need to improve in the manuscript as follows.
- What is “And” in the list of author-name?
Response: We are very sorry for our negligence. We have revised it on line 6 of the manuscript.
- Why some texts were bold (e.g., line 14, line 39-40, line 44-59) whether it is technical problem or intentional. Please correct them.
Response: We have carefully re-checked the full manuscript and corrected any accidental bold fonts.
- Line 41 Reference for statistics of the annual production of aquaculture products should be from the original sources like the Department of Fisheries or FAO. The author may try the software “FishStatJ” to acquire this information. It would provide you with the better recent information on soft-shell turtle production rather than citing the previous article.
Response: We appreciate your suggestions. We have referenced the latest data of China Fisheries Statistics Yearbook in 2021 from the Department of Fisheries and the relevant contents have been revised on line 51 and line 418.
- Line 60-61 I didn’t agree with the authors to mention that “whether these candidates or their homologues are gender-dimorphic ex-60 pressed and present the crucial effects in sex differentiation of P. sinensis, are not well clarified.” I think at least Dmrt1 was very well characterized and proven its function in the sex determination of P. sinensis (Reference no. 25). How it was not well clarified. The author should say the way to add more information rather than saying the way of no information available.
Response: It is really true as reviewer suggested that we have changed this sentence into “Currently, although researchers have conducted relevant studies and achieved some success and achieved some success [16, 17], they are not sufficient to completely reveal the mechanisms of sex determination and sex differentiation in P. sinensis. ” on line 62 and “Therefore, more information of these candidate or their homologues should be provided to help us understand their potential roles in the sex differentiation of P. sinensis.” on line 75.
- Sun, W.; Cai, H.; Zhang, G.; Zhang, H.; Bao, H.; Wang, L.; Ye, J.; Qian, G.; Ge, C. Dmrt1 is required for primary male sexual differentiation in Chinese soft-shelled turtle Pelodiscus sinensis. Sci. Rep. 2017, 7, 4433.
- Zhang, Y.; Xiao, L.; Sun, W.; Li, P.; Zhou, Y.; Qian, G.; Ge, C. Knockdown of R-spondin1 leads to partial sex reversal in genetic female Chinese soft-shelled turtle Pelodiscus sinensis. Gen. Comp. Endocrinol. 2021, 309, 113788.
- Also, the flow of the introduction is not well. Please consider rewriting or rearranging the story for smoother reading.
Response: Considering the reviewer’s suggestion, we have rewritten this story for smoother reading on line 55, 62, 68, 75, and 99.
- Line 69 MHM for the first time present, it needs to be clearer to say that MHM is a Z sex chromosome-linked locus adjacent to the DMRT1 gene.
Response: We are grateful for your valuable suggestions. We have added the sentence of “MHM (Male HyperMethylated), a Z sex chromosome-linked locus adjacent to dmrt1, was methylated and transcriptionally silent in male cells (ZZ), …” on line 87.
- Line 96 Sample size is too small to validate using qPCR primers. Since the author has already developed the primers for qPCR, the author should perform additional samples to assure that these genes were sex differences in other populations as well.
Response: We are grateful for your valuable suggestions. We performed 3 additional biological replicates, bringing the total to 6 for assuring that these genes were sex differences in other populations as well. The qPCR data was showed in Table S9 and the Figure 3 was modified and re-provided based on qPCR results.
- Line 112 Because most Illumina sequencing was performed by outsourcing or company, normally we mentioned the location of sequencing or Illumina equipment. In this case, the author may mention again if the sequencing was performed in your institute (provide the location of sequencing).
Response: Thanks for your suggestions. We have provided the company and location of sequencing, as well as the details of the Illumina equipment on line 133 to line 135.
- All figures were hard to see what it is. The reviewer sees solely that there are differences but is not sure what it is. Please improve the quality of your figures and provide sufficient information into each figure legend.
Response: As Reviewer suggested that we have re-provided all the high quality and sufficient information figures.
- Line 208 The author should provide the correlation coefficient of the results showing that the data of qRT-PCR were consistent with the RNA-seq data.
Response: Thank you for your useful comments and suggestions. The correlation coefficient of the results was showed in Table S9.
- In the discussion part, several lines were redundant to the previous session (e.g., abstract, introduction, and result). For example, line 270-274 was redundant to line 178. Please avoid redundant information.
Response: Special thanks for your comments. We have rechecked the discussion part and deleted the unnecessary redundant information.
- The reviewer suggests the author discuss more novel genes related to sex determination e.g. bmp2, sox3, dkkl1,spo11, and ran in other species. At the current, it was too little to be discussed. The reviewer cannot see how these genes will be important to sex differentiation. The same to discuss ncRNA, circRNA, miRNA, and ceRNA including their interactions. The author should focus to discuss your results and how they could be important for sex differentiation and should be further studied.
Response: Thanks for your advices. We have added the discussion of novel genes and ncRNAs according to the reviewers' comments to focus on the results and how they could be important for sex differentiation on line 298 to line 322 and line 356 to line 370.
Reviewer 2 Report
The authors have tried to identify transcriptional difference between male and female Chinese soft-shelled turtle, mostly surrounding genes related to sex differentiation. The authors should take care that -
1) The quality/resolution of the figures is not good.
2) In the network construction section (3.5) the authors should describe and interpret in a little detail the results they have shown and how the results are relevant.
3) Line 268 - the ncRNA are considered as junk RNA is not correct as its been for long the role of ncRNA is out there in literature. They are well known to play important roles in different forms of regulation.
4) The authors have not shown a PCC or some kind of correlation to show that their 9 replicates for each datasets match.
Author Response
Dear Editors:
On behalf of my co-authors, we thank you very much for giving us an opportunity to revise our manuscript (ID: biology-1721182), we appreciate editor and reviewers very much for their positive and constructive comments and suggestions on our manuscript. We have studied reviewer’s comments carefully and have made revision. We have tried our best to revise our manuscript according to the comments. Attached please find the revised version, which we would like to submit for your kind consideration.
We would like to express our great appreciation to you and reviewers for comments on our paper. Looking forward to hearing from you.
Thank you and best regards.
Yours sincerely,
Corresponding author: Xinping Zhu
E-mail: zhuxinping_1964@163.com
Reviewer #2:
Comments and Suggestions for Authors
The authors have tried to identify transcriptional difference between male and female Chinese soft-shelled turtle, mostly surrounding genes related to sex differentiation. The authors should take care that –
- The quality/resolution of the figures is not good.
Response: As Reviewer suggested that we have re-provided high quality/resolution of all figures.
- In the network construction section (3.5) the authors should describe and interpret in a little detail the results they have shown and how the results are relevant.
Response: We appreciate your suggestions. The miRNAs interaction with lncRNAs, circRNAs and mRNAs using miRanda software with default parameters [1]. The fold change greater than 2 and less than -2 indicated up and down regulation, and q value <0.05. Cytoscape [2] was used for visualization the network. And the details of the results were added on line 250 to line 253.
- Enright, A. J.; John, B.; Gaul, U.; Tuschl, T.; Sander, C.; Marks, D. S. MicroRNA targets in Drosophila. Genome Biol. 2003, 5, R1.
- Shannon, P.; Markiel, A.; Ozier, O.; Baliga, N. S.; Wang, J. T.; Ramage, D.; Amin, N.; Schwikowski, B.; Ideker, T. Cytoscape: a software environment for integrated models of biomolecular interaction networks. Genome Res. 2003, 13, 2498-504.
- Line 268 - the ncRNA are considered as junk RNA is not correct as its been for long the role of ncRNA is out there in literature. They are well known to play important roles in different forms of regulation.
Response: Thank you for your useful comments and suggestions. It is really true as reviewer suggested that we have deleted this part according to the reviewers' comments on line 327.
- The authors have not shown a PCC or some kind of correlation to show that their 9 replicates for each datasets match.
Response: We are grateful for your valuable suggestions. We have provided the Pearson's correlation coefficient of each sample showed in Figure S2.
Figure S2. Pearson's correlation coefficient of each sample

Reviewer 3 Report
Minor editorial revisions will be needed.
- line 6: and* Is the last author missing?
- line 69: MHM should be spelled out: MHM ()
- line 74: AGO2 should be spelled out: AGO2 ()
- lines 99, 100, 112, 113: add city and state before USA, or company name, city, state, country after equipment used
- lines 144, 145: add the company name, city, state and country after Software
- line 152: delete % after 45.70 45.99
- lines 212, 212-213: add commas before and after respectively
- line 345: missing volume and pages
Author Response
Dear Editors:
On behalf of my co-authors, we thank you very much for giving us an opportunity to revise our manuscript (ID: biology-1721182), we appreciate editor and reviewers very much for their positive and constructive comments and suggestions on our manuscript. We have studied reviewer’s comments carefully and have made revision. We have tried our best to revise our manuscript according to the comments. Attached please find the revised version, which we would like to submit for your kind consideration.
We would like to express our great appreciation to you and reviewers for comments on our paper. Looking forward to hearing from you.
Thank you and best regards.
Yours sincerely,
Corresponding author: Xinping Zhu
E-mail: zhuxinping_1964@163.com
Reviewer #3:
Comments and Suggestions for Authors
Minor editorial revisions will be needed.
- line 6: and* Is the last author missing?
Response: We are very sorry for our negligence. We have revised it on line 6 of the manuscript.
- line 69: MHM should be spelled out: MHM ()
Response: We appreciate your suggestions. We have spelled out the full name of MHM (Male HyperMethylated) on line 87.
- line 74: AGO2 should be spelled out: AGO2 ()
Response: We are grateful for your valuable suggestions. We have spelled out the full name of AGO2 (Argonaute 2) on line 93.
- lines 99, 100, 112, 113: add city and state before USA, or company name, city, state, country after equipment used
Response: We are grateful for your valuable suggestions. We have added the city and state before USA and the company name, city, state, country after equipment used on line 119 to 121, line 230 and line 127 to 135.
- lines 144, 145: add the company name, city, state and country after Software
Response: Thanks for your valuable comments. We have added the company name, city, state and country after Software on line 166 to 169.
- line 152: delete % after 45.70 45.99
Response: We appreciate your advice. We have deleted % after 45.70 and 45.99 on line 176.
- lines 212, 212-213: add commas before and after respectively
Response: According to the reviewers' comments, we have added commas before and after respectively on line 239 to 241.
- line 345: missing volume and pages
Response: We are very sorry for our negligence. We have added the missing volume and pages on line 425.

Round 2
Reviewer 1 Report
The current version of the manuscript is much improved than the previous one and good for publication. Thanks to the authors for making the corrections and comprehensively discussing star genes in turtles' sex determination and differentiation.
Reviewer 2 Report
Accept as it is!